# Soyasapogenol-B as a Potential Multitarget Therapeutic Agent for Neurodegenerative Disorders: Molecular Docking and Dynamics Study

**DOI:** 10.3390/e24050593

**Published:** 2022-04-23

**Authors:** Danish Iqbal, Syed Mohd Danish Rizvi, Md Tabish Rehman, M. Salman Khan, Abdulaziz Bin Dukhyil, Mohamed F. AlAjmi, Bader Mohammed Alshehri, Saeed Banawas, Qamar Zia, Mohammed Alsaweed, Yahya Madkhali, Suliman A. Alsagaby, Wael Alturaiki

**Affiliations:** 1Department of Medical Laboratory Sciences, College of Applied Medical Sciences, Majmaah University, Majmaah 11952, Saudi Arabia; a.dukhyil@mu.edu.sa (A.B.D.); b.alshehri@mu.edu.sa (B.M.A.); s.banawas@mu.edu.sa (S.B.); qamarzia@mu.edu.sa (Q.Z.); m.alsaweed@mu.edu.sa (M.A.); y.madkhali@mu.edu.sa (Y.M.); s.alsaqaby@mu.edu.sa (S.A.A.); w.alturaiki@mu.edu.sa (W.A.); 2Health and Basic Sciences Research Center, Majmaah University, Al Majmaah 15341, Saudi Arabia; 3Department of Pharmaceutics, College of Pharmacy, University of Hail, Hail 81442, Saudi Arabia; 4Department of Pharmacognosy, College of Pharmacy, King Saud University, Riyadh 11451, Saudi Arabia; mrehman@ksu.edu.sa (M.T.R.); malajmii@ksu.edu.sa (M.F.A.); 5Clinical Biochemistry & Natural Product Research Laboratory, Department of Biosciences, Integral University, Lucknow 226026, U.P., India; contactskhan@gmail.com; 6Department of Biomedical Sciences, Oregon State University, Corvallis, OR 97331, USA

**Keywords:** Alzheimer’s Disease, molecular docking, molecular dynamics simulations, neurological disorders, Soyasapogenol B

## Abstract

Neurodegenerative disorders involve various pathophysiological pathways, and finding a solution for these issues is still an uphill task for the scientific community. In the present study, a combination of molecular docking and dynamics approaches was applied to target different pathways leading to neurodegenerative disorders such as Alzheimer’s disease. Initially, abrineurin natural inducers were screened using physicochemical properties and toxicity assessment. Out of five screened compounds, a pentacyclic triterpenoid, i.e., Soyasapogenol B appeared to be the most promising after molecular docking and simulation analysis. Soyasapogenol B showed low TPSA (60.69), high absorption (82.6%), no Lipinski rule violation, and no toxicity. Docking interaction analysis revealed that Soyasapogenol B bound effectively to all of the targeted proteins (AChE, BuChE MAO-A, MAO-B, GSK3β, and NMDA), in contrast to other screened abrineurin natural inducers and inhibitors. Importantly, Soyasapogenol B bound to active site residues of the targeted proteins in a similar pattern to the native ligand inhibitor. Further, 100 ns molecular dynamics simulations analysis showed that Soyasapogenol B formed stable complexes against all of the targeted proteins. RMSD analysis showed that the Soyasapogenol B–protein complex exhibited average RMSD values of 1.94 Å, 2.11 Å, 5.07 Å, 2.56 Å, 3.83 Å and 4.07 Å. Furthermore, the RMSF analysis and secondary structure analysis also indicated the stability of the Soyasapogenol B–protein complexes.

## 1. Introduction

Alzheimer’s disease (AD) is the most common form of dementia, characterized by a gradual deterioration in cognitive abilities [1]. Pathological features of the condition include amyloid-beta (Aβ) and tangled intracellular neurofibrils with aggregated tau protein (hyperphosphorylated), as well as synaptic and neuronal loss [2,3,4]. Currently, there are more than 55 million people affected by AD globally, with an increase of 10 million new cases every year, which is predicted to grow to 106.4 million by 2050 [5]. The rapidly increasing prevalence of AD is likely to put a major burden on families and the community [1].

To explore the various critical factors involved in AD, potential targets have been taken into consideration. Glycogen synthase kinase 3 (GSK3β) is a proline-directed kinase that controls various physiological processes, including glycogen metabolism and gene transcription. GSK3β is vital in AD pathogenesis, both in sporadic and familial forms [6]. According to the cholinergic hypothesis, AD symptoms are predominantly attributed to structural changes in cholinergic synapses, the loss of particular subtypes of acetylcholine (ACh) receptors, the death of ACh-generating neurons, and, as a result, cholinergic neurotransmission degradation. As a consequence of these problems, the ACh-hydrolyzing enzyme acetylcholinesterase (AChE) accumulates [7]. BuChE is an intrinsic bio scavenger that functions as the first line of protection toward toxic substances, which could suppress AChE activity. Furthermore, BuChE may compensate for AChE and maintain the cortical ACh level; mutations in the BuChE gene that decrease enzyme activity may postpone disease development in patients at risk of AD [8].

The excitatory glutamatergic neurotransmission mediated by the N-methyl-d-aspartate receptor (NMDAR) is required for synaptic plasticity and neuron survival. Increased NMDAR activity, on the other hand, induces excitotoxicity and encourages cell death, suggesting a possible neurodegeneration mechanism in AD. The oxidative deamination of biogenic and xenobiotic amines is facilitated by monoamine oxidase (MAO), which plays a significant role in the metabolism of neuroactive and vasoactive amines in the CNS and peripheral tissues. In several main physiological mechanisms, studies have found that MAO participation in neurological disorders and neurodegenerative diseases is significant. MAO-B has been suggested as a biomarker and activated MAO-B induces cognitive dysfunction, kills cholinergic neurons, disrupts the cholinergic system, and contributes to the formation of amyloid plaques [9]. Despite recent advancements over three decades, still there is no cure for AD. As brain-derived neurotrophic factor (BDNF) or abrineurin has a critical role in remodeling the cognitive functions and remodeling of synapses, restoring the BDNF expression could be a therapeutic pathway [10]. Meanwhile, a variety of phytochemicals and natural products, like compounds, have been reported as potent antidiabetic, antimicrobial, antioxidant, gastro-protection, geno-protective, hypolipidemic and neuroprotective agents [11,12,13,14,15,16,17,18,19,20,21,22,23,24,25,26]. Studies have also shown good improvements in neurodegenerative diseases through the use of phytochemicals [27], and indicating that phytochemicals, which are chemical compounds extracted from plants, exhibit possible health promotion properties. Therefore, we focused on exploring natural phytochemicals having properties as BDNF inducers for inhibiting the various protein targets involved in AD. Hence, the 44 natural phytochemicals emphasized as BDNF inducers were selected for the study to assess their multitargeting potential against the target proteins AChE, BuChE, GSK3β, MOA-A, MOA-B, and NDMA as a potential molecule for treating AD. After computational screening, Soy-B (Soyasapogenol B) was predicted to be the most promising candidate against different targets of AD. BDNF induction could serve as one of the therapeutic strategies against neurological disorders; thus, compounds based on BDNF inducer scaffold might provide a plausible multi-target agent against AD. It is noteworthy to mention that this is a first report where Soy-B has been explored for its multi-targeting ability against AD. However, experimental validation is required to confirm our preliminary findings. Nevertheless, the preliminary findings of the study provided some important insights of Soy-B interactions with different targets of AD that might help in designing more effective and potent agent against AD.

## 2. Results

### 2.1. Physicochemical Properties and Prediction of Toxicity Potential of BDNF Inducers

Natural compounds’ physicochemical properties and ADMET properties were evaluated for drug likeness using the Orisis Datawarrior property explorer tool. Of the 44 selected natural BDNF inducers, Hericene A, Physodic acid, Perlatolic acid, Ginsenosides Rg1, Ginsenosides Rb1, Salvianolic acid B, Bacoside A Chemspider, Crocin, Fucoxanthin, Hyperforin, Hypericin, Aconitine, Hypaconitine, Mesaconitine (total 14 compounds) violated Lipinski’s rule, and only 18 compounds were found to have less than 90 Å TPSA (required to cross blood–brain barrier) and better (more than 70%) gastro-intestinal absorption (Appendix A). We further analyzed the toxicity potential of 30 BDNF inducers (those that did not violate Lipinski’s rule), and found that only 13 compounds exhibited no toxicity for four parameters: mutagenic, tumorigenic, reproductive effect, and irritant. Among these 13 non-toxic compounds, we found that only five compounds were among those 18 compounds that were selected earlier on the basis of their intestinal absorption percentage and lower TPSA, suggesting their efficiency in crossing the blood–brain barrier. Soyasapogenol B, Huperzine A, Hydroxytyrosol, Alpinetin, and Calycosin showed good absorption efficiency without violating the Lipinski rule (Appendix A), with minimal or no toxicity (Appendix A).

### 2.2. Molecular Docking Analysis

The five selected compounds (Soyasapogenol B, Huperzine A, Hydroxytyrosol, Alpinetin and Calycosin) and five reference inhibitors (Tacrine, ADP, Harmine. Safinamide, Dichlorokynurenic acid) were subjected to molecular docking using AutoDock Vina. All of the compounds were docked with different neurological target proteins (AChE, BuChE, GSK3β, MOA-A, MOA-B, NDMA) (Table 1).

Based on the Binding Energy (ΔG), Soyasapogenol B (Soy-B) showed a better binding energy score, of about −8.7 kcal/mol, −9.9 kcal/mol, −8.7 kcal/mol, −7.9 kcal/mol, −7.6 kcal/mol and −8.4 kcal/mol, than all other BDNF inducers against AChE, BuChE, GSK3β, MOA-A, MOA-B and NDMA, respectively. Tacrine exhibited a binding affinity of −7.4 kcal/mol and −8 kcal/mol against AChE and BuChE, respectively. ADP and Dichlorokynurenic acid showed a binding affinity of about −6.7 kcal/mol and −6.9 kcal/mol against GSK3β and NDMA receptor, respectively, whereas Harmine exhibited a binding affinity of −6.5 kcal/mol for MOA-A, and Safinamide showed binding score of −5.8 kcal/mol towards MOA-B.

#### 2.2.1. Analysis of Interaction of Soy-B with AChE

In the study, interaction between Soy-B and AChE revealed that Soy-B interacted with the residues close to the catalytic active site of AChE, and hence showed a non-competitive inhibition (Figure 1). The binding mode of Soy-B was further compared with native control ligand (Tacrine), and the protocol was standardized by re-docking the native ligand (Figure 1B). Native ligand in the co-crystalized structure interacted with TRP84 and PHE330 amino acid residues of AChE via pi–pi stacking and pi–alkyl interactions, respectively. Similar results were observed when native ligand was redocked with AChE (Figure 1E). Moreover, eleven amino acid residues (SER 81, GLY117, GLY118, GLU199, SER200, TYR 334, TRP432, ILE439, HIS440, GLY441, TYR442) of AChE were involved in van der Waals’ interactions with redocked ligand. On the other hand, Soy-B interacted with AChE near to the native ligand binding site (Figure 1C) through two strong hydrogen bonds via ASP285 and GLY335 amino acid residues of AChE. Stability of Soy-B and AChE binding was further enhanced by van der Waals’ interactions with TYR70, ASP276, VAL277, TRP279, ASN280, SER286, ILE287, TYR334 and LEU358 amino acid residues (Figure 1D).

#### 2.2.2. Analysis of Interaction of Soy-B with BuChE

Interaction between Soy-B and BuChE revealed that Soy-B binds within the active site cavity of BuChE (Figure 2). Comparative analysis of Soy-B binding mode was done with native control ligand (Tacrine) and standardization was performed by re-docking the native ligand (Figure 2C). Hydrophobic pi–pi stacking and pi–alkyl interactions were observed during native ligand binding with BuChE active site, and similar interactions were found when the native ligand was redocked with BuChE (Figure 2C,E). Not only hydrophobic interactions, but also van der Waals’ interactions played an important role in binding of native ligand to the BuChE active site. However, Soy-B showed only van der Waals’ interactions via 17 amino acid residues (ASN68, ASP70, GLY78, SER79, TRP82, GLY116, GLY117, GLN119, THR120, PRO285, SER287, ASN289, ALA328, PHE329, TYR332, TRP430, HIS438) of BuChE (Figure 2D).

#### 2.2.3. Analysis of Interaction of Soy-B with GS3Kβ

Soy-B interaction with GS3Kβ is shown in Figure 3. The binding mode of Soy-B interaction was compared with native ligand control, ADP (Figure 3B). In this study, interaction GS3Kβ inhibitor bromoindirubin was also observed. Interestingly, all of the tested compounds, inhibitors and native ligands bound to the same position in the active site cavity of GS3Kβ (Figure 3B). Superimposition of the native ligand, the redocked native ligand, and the inhibitor at the active site confirmed the standardization of the protocol (Figure 3C,D). However, hydrogen bonds, electrostatic and van der Waals’ interactions dominated the stability of native ligand and GS3Kβ binding (Figure 3F). On the other hand, Soy-B showed one hydrogen bond with GS3Kβ through GLN185 amino acid residue and one electrostatic interaction with LYS183, followed by seventeen hydrophobic interactions (Figure 3E).

#### 2.2.4. Analysis of Interaction of Soy-B with MOA-A

Interaction between Soy-B and MOA-A is shown in Figure 4. Binding mode of Soy-B interaction with MOA-A was compared with the native ligand control, Harmine (Figure 4A,B). All of the tested compounds and the native ligand bound to the same position in the active site cavity of MOA-A (Figure 4B). Superimposition of the native ligand and the redocked native ligand at the active site confirmed the standardization of the protocol (Figure 4C). Redocking of the native ligand (Figure 4E) showed hydrogen bonds and hydrophobic interactions, whereas Soy-B binding with MOA-A showed two hydrogen bonds with CYS323 and THR336, one electrostatic interaction with PHE352, and 19 hydrophobic interactions (Figure 4D).

#### 2.2.5. Analysis of Interaction of Soy-B with MOA-B

All five of the selected natural inducers and redocked native ligand interacted with the same catalytic active site pocket of MOA-B (Figure 5A,B). The interaction pattern of the best hit (Soyasapogenol B) with the target protein is represented in Figure 5. The protocol was validated through redocking of the native ligand, and it was observed that redocked ligand occupied the same residues (LEU171, CYS172, ILE199, GLN206, ILE316, TYR326) as the native ligand (Figure 5C,E). It was also observed that Soy-B and MOA-B complexes were stabilized by three (LEU171, ILE198, TYR398) important residues of the catalytic site. Moreover, several van der Waals’ interactions were also observed in stabilizing the Soy-B and MOA-B complexes (Figure 5D).

#### 2.2.6. Analysis of Interaction of Soy-B with NMDA

The interaction binding pattern of compounds and native ligands revealed that all of the ligands occupied the active site gorge residues (Figure 6A–C). Redocked native ligand bound to the same location as of co-crystallized native ligand in NMDA with hydrogen bond (PRO124, THR126, ARG131), electrostatic interaction (ASP224), hydrophobic interactions (PHE16, PHE92, PRO124, VAL227, PHE250), and several van der Waals’ interactions (Figure 6E). The complex of Soy-B and NMDA was stabilized mostly by the van der Waals’ interactions and one residue (THR126) were observed to interacted with hydrogen bond (Figure 6D).

### 2.3. Molecular Dynamic (MD) Simulation Analysis

#### 2.3.1. Root Mean Square Deviation (RMSD) Analysis

MD simulation analysis was carried out to understand the selected proteins and Soy-B complexes’ nature of interaction (dynamic) and stability. Figure 7 presents the MD simulation of Protein-Soy-B complex for 100 ns. The root-mean-square deviation (RMSD) is a measure that accounts for the protein stability and its deviation from the initial structure during the MD simulation [28]. For the early 10 ns, the AChE, RMSD alone ranged between 0.00 and 2.04 Å. Then, the subsequent simulation was constant. It showed that the AChE structure during the simulation was not considerably altered. For the first 10 ns, and then 0.50 Å for the rest of the simulation, the AChE-Soy-B complex’s RMSD varied within 0.00–2.96 Å. The average RMSD values of AChE alone and in complex with Soy-B during 10–100 ns simulation were estimated as 2.18 and 1.94 Å, respectively (Figure 7a).

Similarly, the RMSD of BuChE alone and bound with Soy-B remained consistent during the whole simulation. The average RMSD values of BuChE alone and in combination with Soy-B were determined to be 2.03 and 2.11 Å, respectively (Figure 7b). Likewise, the RMSD value of MAO-A alone fluctuated during 0–10 ns and then remained constant, with an average value of 4.38 Å. The RMSD of the MAO-A and Soy-B complex remained consistent, with an average value of 5.07 Å (Figure 7c).

Similarly, the RMSD of MAO-B in the absence and presence of Soy-B fluctuated for the initial 0–10 and 0–35 ns, respectively, and then acquired a stable configuration throughout the remainder of the simulation. The average RMSD values of MAO-B alone and in complex with Soy-B during 30–100 ns of the simulation were estimated at 2.34 and 2.56 Å, respectively (Figure 7d). In addition, the RMSD of GS3Kβ in the absence and presence of Soy-B fluctuated for the initial 0–10 ns and then acquired a stable configuration throughout the remainder of the simulation. The average RMSD values of GS3Kβ alone and in complex with Soy-B during 30–100 ns simulation were estimated at 3.54 and 3.83 Å, respectively (Figure 7e). Likewise, the RMSD value of NMDA alone fluctuated during 0–10 ns and then remained constant with an average value of 3.83 Å. The RMSD of the NMDA and Soy-B complex remained consistent, with an average value of 4.07 Å (Figure 7f).

None of the RMSD fluctuations exceeded the allowed limit of 2.0 Å. These results showed that the Soy-B combination with AChE, BuChE MAO-A, MAO-B, GS3Kβ, and NMDA established stable complexes.

#### 2.3.2. Root Mean Square Fluctuation (RMSF) Analysis

Figure 8 depicts the variation in RMSF values of Soy-B bound with AChE, BuChE MAO-A, MAO-B, GS3Kβ and NMDA. The residues showing higher peaks correspond to loop regions or N and C-terminal zones. The RMSF of AChE, BuChE MAO-A, MAO-B, GS3Kβ and NMDA did not deviate significantly in the presence of Soy-B, ensuring that the overall conformation of the corresponding proteins remained conserved.

#### 2.3.3. Analysis of Radius of Gyration (Rg) and Solvent-Accessible Surface Area (SASA)

The variation in Rg of Soy-B bound with different proteins (AChE, BuChE, MAO-A, and MAO-B) as a function of simulation time is presented in Figure 9a. The results show that Rg of different protein–ligand systems fluctuated within the acceptable limit throughout the simulation. The average Rg values of AChE-Soy-B and BuChE-Soy-B complexes were estimated as 4.55 and 4.60 Å, respectively. Conversely, the average Rg values of MAO-A-Soy-B and MAO-B-Soy-B complexes were determined to be 4.56 and 4.57 Å, respectively (Figure 9a). The solvent-accessible surface area (SASA) of Soy-B bound to AChE varied significantly during 0–15 ns of simulation and remained constant for the rest of the simulation (Figure 9b). The SASA of BuChE-Soy-B and MAO-A-Soy-B complexes varied slightly within the acceptable limits, whereas the SASA of MAO-B-Soy-B complex fluctuated during 20–35 ns of simulation. The SASA of all protein–ligand complexes remained consistent for a large part of the simulation, during 40–100 ns. The average SASA values of Soy-B bound with AChE, BuChE, MAO-A, and MAO-B during 40–100 ns of the simulation were 210.7, 65.9, 193.5 and 372.1 Å2, respectively (Figure 9b).

The average Rg values of GS3Kβ-Soy-B and NMDA-Soy-B complexes were estimated at 4.59 and 4.60 Å, respectively (Figure 9c). The solvent-accessible surface area (SASA) of the complexes did not vary, and remained constant for the rest of the simulation. The average SASA values of Soy-B bound with GS3Kβ and NMDA B during 100 ns of the simulation were 160.1 and 171.2 Å^2^, respectively (Figure 9d). All of these results suggest that Soy-B remained inside the binding cavity of AChE, BuChE, MAO-A, MAO-B, GS3Kβ and NMDA in a stable conformation.

#### 2.3.4. Secondary Structure Analysis

The interaction between a ligand and protein often leads to changes in a protein’s secondary structural elements (SSE). Thus, a check on the variation in SSE during simulation is critical to overview establishing a stable complex between Soy-B and targeted proteins. The variation in total SSE (α-helix + β-sheet) of proteins (AChE, BuChE, MAO-A, MAO-B, GS3Kβ and NMDA) bound with Soy-B during simulation is presented in (Figure 10). We found that the total SSE of AChE, BuChE, MAO-A, MAO-B, GS3Kβ and NMDA in complex with Soy-B was 38.7% (α-helix: 25.7% and β-sheets: 13.0%), 39.8% (α-helix: 26.8% and β-sheets: 13.0%), 41.4% (α-helix: 26.0% and β-sheets: 15.4%), 41.1% (α-helix: 25.3% and β-sheets: 15.8%), 42.1% (α-helix: 26.2% and β-sheets: 15.9%), and 34.7 % (α-helix: 21.4% and β-sheets: 13.3%), respectively. It should be noted that the SSE of all of the targeted proteins in combination with Soy-B remained consistent throughout the simulation, suggesting a stable interaction between proteins and ligand.

#### 2.3.5. Protein–Ligand Interaction Analysis

As seen in Figure 11, hydrogen bonds and hydrophobic interactions play an essential role in sustaining and stabilizing the complexes of targeted proteins and Soy-B. In addition, water bridges and ionic association are also critical to forming a stable protein–ligand complex. The amino acid residues SER235, GLU306, and ASN525, were the most important in forming hydrogen bonds between AChE and Soy-B (Figure 11a). Similarly, amino acid residues such as ASN68, TRP82, and ASN289 were observed to stabilize the BuChE-Soy-B complex through hydrogen bonding, while PRO285, ALA328, PHE329 and TYR332 were involved in hydrophobic interactions (Figure 11b). Likewise, in the formation of the MAO-A and Soy-B complex, amino acid residues such as ASN125, ASN212, and GLU485 were involved in hydrogen bond formation, while ALA111, TYR124, and TRP128 were engaged with Soy-B through hydrophobic interactions (Figure 11c). Moreover, in MAO-B and Soy-B complex formation, amino acid residue ARG307 was predominantly involved in hydrogen bond formation, and TYR80 formed hydrophobic interaction (Figure 11d). In addition, GS3Kβ and Soy-B complex formed amino acid residues VAL135 and LYS183 hydrogen bond formation while ALA83, LEU132, and LEU188 formed hydrophobic interactions (Figure 11e). Moreover, in NMDA and Soy-B complex formation, amino acid residue SER248, PHE246, ASP224, and GLN144 were predominantly involved in hydrogen bond formation, and TYR184, ILE183, and LEU146 formed hydrophobic interactions (Figure 11f).

The involvement of amino acid residues in terms of percent simulation time during the formation of a complex between Soy-B and targeted proteins (AChE, BuChE, MAO-A, MAO-B, GS3Kβ and NMDA) was also established (Figure 12). Only those amino acid residues which interacted with ligand for more than 30% simulation time are shown. It is clear that during the simulation of ACHE-Soy-B, the amino acid residues SER235, LEU305, and ASN525 were involved with the ligand for about 36%, 43% and 41% of simulation time, respectively (Figure 12a). Similarly, during the BuChE-Soy-B simulation, TRP82 and ASN289 were engaged for 96% and 68%, respectively (Figure 12b). Likewise, ASN125 of MAO-A interacted with Soy-B for about 45% of simulation time through a water bridge (Figure 12c), while ARG307 of MAO-B formed two interactions with two different hydroxyl groups of Soy-B for 59% and 35%, respectively (Figure 12d). Soy-B formed interactions with GS3Kβ at the VAL135 amino residue, forming hydroxyl bonds for about 76% of the simulation (Figure 12e), whereas with NMDA, hydroxyl interactions were formed with the target proteins PHE246, SER248, and ASP224, which were stable throughout 94%, 87% and 40% of the simulation time, respectively.

In addition, the formation of a stable protein and ligand complex was established by determining the total number of contacts formed between them during simulation (Figure 13). It is clear that during the simulation, the total number of contacts between Soy-B and AChE, BuChE, MAO-A, MAO-B, GS3Kβ, and NMDA varied between 0 and 8, 2 and 9, 0 and 9, 0 and 7, 2 and 9, and 3 and 9, respectively. On average, AChE, BuChE, MAO-A, MAO-B, GS3Kβ and NMDA formed 5, 5, 4, 4, 5 and 5 contacts with Soy-B.

Thus, from the results it can be inferred that the Soy-B was shown to have good affinity, and tends to form stable complexes with hydroxyl group interactions.

## 3. Discussion

Alzheimer’s Disease (AD) is a neurodegenerative disease affecting 55 million people globally, which is expected to increase over 106.4 million by 2050 [5,29]. Even though there are tremendous research efforts directed toward AD treatment, an effective drug is still lacking. Therefore, there is a need for alternate strategies for AD treatment. Since phytochemicals remain a partially explored pool of therapeutic strategies, the present study was intended to virtually screen the 44 natural BDNF inducers against the AChE, BuChE, GSK3β, MOAA, MOAB, NDMA proteins to identify potential therapeutic candidates for AD treatment. AChE is a key enzyme of the cholinergic system, and has been reported to be deteriorated in the progression of AD, which is accompanied by the decline of acetylcholine and the loss of cholinergic neurons in the forebrain. There are multiple therapeutic modalities targeting the cholinergic system; in particular, AChE is the main target that has been proven to be a promising therapeutic strategy [30]. BuChE is a serine hydrolase enzyme that plays a crucial role in the regulation of acetylcholine metabolism, and is closely related to AChE. It tends to be predominantly expressed in the white matter, glia and neurons regulating cognition and behavior in the AD condition [31]. In an AD mouse model, accumulation of fibrillary Aβ plaques indicated an active role in the pathological development of AD. Using a strong and specific BuChE inhibitor to increase acetylcholine levels and intimidate fibrillar Aβ accumulation has been demonstrated to be an easy and successful treatment strategy for AD [32].

Both Monoamine oxidase-A (MAO-A) and MAO-B are involved in AD pathogenesis. Neurodegeneration in AD is related to the altered function of any isoform or any associated disturbance of neurotransmitter substrates, like serotonin, dopamine, or noradrenaline. MAO might possibly play a role in neuropathology due to the production of hydrogen peroxide as a byproduct of the deamination reaction. When antioxidant mechanisms are weakened, such as during aging, and especially in AD, the resulting oxidative stress could lead to cell death, which inevitably involves the role of mitochondria. MAO-A-immunoreactive cell loss is amplified in late-stage cognitive impairment in brainstem monoaminergic nuclei and other locations. MAO-B is predominantly expressed in glia and has been thoroughly investigated with respect to its function in neurodegeneration. MAO-A/B-associated shift in AD levels of aminergic neurotransmitters likely contributes to the neurobiology of a spectrum of neurological disorders in AD populations [33]. GSK3β leads to tau protein hyperphosphorylation, the major component of neurofibrillary tangles (NFTs), which are hallmarks of AD. In addition, GSK3β regulates several neuronal functions, such as Aβ-induced cell death, axonal transportation, cholinergic functioning, and adult neurogenesis that are dysregularized throughout the pathogenesis of AD [34]. NMDA plays a key role in the synaptic transmission and plasticity of learning, memory, and the development of the nervous system, as well as in neurotoxicity. Ca2+ signaling causes neuronal cell death, which is clinically linked to cognitive impairment and the establishment of pathological architecture in AD. This also justifies the clinical trial of the NMDA receptor antipathist memantine as a protective and symptomatologic treatment for AD [35].

A total of 44 natural BDNF inducer compounds were assessed for their toxicity, and the physicochemical properties of all of these compounds were evaluated. TPSA was analyzed in order to assess the efficiency of crossing BBB, and it was found that Imperatorin, Eugenol, Auraptene, Soyasapogenol B (Soy-B), Tanshinone II-A, Caffeine, Honokiol and Magnolol possessed TPSA < 60. The BBB is a characteristic barrier that limits the efficiency of therapeutic modalities by reducing their bioavailability as a result of their inability to cross the BBB [36]. Therefore, crossing the BBB is crucial in treating neurodegenerative disorders. Cumulative analysis of the Lipinski rule and toxicity in the BDNF inducers showed that Soy-B, Huperzine A, Hydroxytyrosol, Alpinetin, and Scopoletin demonstrated good absorption efficiency without violating the Lipinski rule or exhibiting toxicity.

Molecular docking analysis suggested that Soy-B had a binding energy score of about −8.7 Kcal/mol, −9.9 Kcal/mol, −8.7 Kcal/mol, −7.9 Kcal/mol, −7.6 Kcal/mol and −8.4 Kcal/mol against the target proteins AChE, BuChE, GSK3β, MOAA, MOAB and NDMA, respectively, on the basis of which Soy-B was selected as a candidate for further simulation studies.

Soy-B interacted with AChE through TYR70, ASP276, VAL277, TRP279, ASN280, ASP285, SER286, ILE287, TYR334, GLY335 and LEU358 amino acid residues (Figure 1D). In another study, semi-synthetic piperidine alkaloids also interacted with the same amino acid residues TYR70, ASP285, SER286, TYR334 and GLY335 of the active site of AChE [37]. On the other hand, the interaction of Soy-B with BuChE showed the involvement of ASN68, ASP70, GLY78, SER79, TRP82, GLY116, GLY117, GLN119, THR120, PRO285, SER287, ASN289, ALA328, PHE329, TYR332, TRP430, and HIS438 amino acid residues (Figure 2D). Similary, all of these amino acids were reported to be involved in the interaction of Bacoside X with BuChE in a recent study [38]. Soy-B exhibited an interaction with the ATP binding pocket of GS3Kβ through GLN185 and LYS183 amino acid residues (Figure 3E). In a similar way, Salvianolic acid B interacted with the same active site pocket of GS3Kβ via the GLN185 and LYS183 amino acids [39]. Moreover, Soy-B binding with MOA-A showed a strong interaction via the CYS323, THR336, and PHE352 amino acid residues (Figure 4D), while Soy-B and MOA-B interaction was stabilized by the LEU171, ILE198, and TYR398 amino acid residues (Figure 5D). Thiol of CYS323 of MOA-A plays a crucial role in interaction with different compounds, such as piperine and β -carboline analogs [40,41]. Importantly, Soy-B also showed interaction with the CYS323 residue of MOA-A. In one study, desmodeleganine (a natural alkaloid) was shown to interact with the LEU171 and ILE198 amino acid residues of MOA-B [42]. Meanwhile, several natural compound derivatives interacted with the aromatic cage of MOA-B via TYR398 [43]. In addition, inhibitor binding and the important role of the TYR398 amino acid residue of MOA-B were also reported in a recent study [44]. Interestingly, Soy-B was shown to interact with all of these crucial amino acid residues of MOA-B. Soy-B and NMDA interaction was stabilized by strong bonding with THR126 (Figure 6D). Similarly, catechin, gingetin and ginkgolic acid showed strong hydrogen bonding with THR126 [45]. Furthermore, MS analysis (RMSD analysis, RMSF analysis, secondary structure, and protein ligand interaction analysis) during 0–100 ns indicated that the Soy-B had formed stable complexes with the targeted proteins (AChE, BuChE MAO-A, MAO-B, GS3Kβ, and NMDA). Figure 7a–f presents the average RMSD value for Soy-B -protein complex for AChE, BuChE, GSK3β, MOAA, MOAB and NDMA as 1.94 Å, 2.11 Å, 5.07 Å, 2.56 Å, 3.83 Å and 4.07 Å, respectively. However, none of the RMSD values fluctuated more than the acceptable limit of 2 Å [46].

Soy-B belongs to the soyasaponins predominantly isolated from Glycine max and shown to exert various pharmacological properties (anti-inflammatory [47], phytooestrogenic [48] and memory enhancement [49,50]). Soy-B has been shown to attenuate LPS-induced memory impairment in mice via NF-κB-mediated BDNF expression and prevent scopolamine-induced memory impairment [50,51]. In LPS-treated mice and corticosterone-stimulated SH-SY5Y cells, Soy-B enhanced cAMP response element-binding protein phosphorylation and BDNF expression, and reduced NF-B activation. Soyasaponin subtypes A1, A2 and I reduced LPS-induced cyclooxygenase 2 (COX-2) production via inhibiting NF-κB. Soyasaponins appear to reduce memory loss by inhibiting NF-B-mediated inflammation, according to this research [52,53]. In the present study, Soy-B was predicted to be the best among all of the compounds screened. However, the limitation of the present study is that our findings are only based on the in silico approach, and need to be further confirmed via in vitro and in vivo experiments.

## 4. Materials and Methods

### 4.1. Calculation of Physicochemical Properties and Prediction of Toxicity Potential

A total of 44 Natural BDNF inducer compounds were assessed for their toxicity, and the physicochemical properties of all of the compounds were assessed using the Orisis Datawarrior property explorer tool [54]. The tool was also used to assess the molecular weight, the number of hydrogen bond donors and acceptors, rotatable bonds, cLogP value, topological polar surface area, and Lipinski’s rule violation [55]. The absorption % was calculated per Zhao et al. [56] using the formula Absorption %=109−(0.345×TPSA). In the Orisis Datawarrior tool’s analysis, the predicted toxicity values were dependent on comparing the precalculated investigated molecules with the structures of the tested molecules. Various aspects and effects of the toxicity, including tumorigenicity, mutagenicity, and irritability of the tested compounds, were also tested using the Orisis Datawarrior tool.

### 4.2. Molecular Docking Analysis of Natural BDNF Inducers against Target Proteins

#### 4.2.1. Target Retrieval and Preparation

The 3-dimensional structure of Alzheimer’s Disease targets, i.e., GSK3b (PDB ID: 1J1C), AChE (PDB ID: 1ACJ), BChE (PDB ID: 4BDS), NMDA (PDB ID: 1PBQ), MAO-A (PDB ID: 2Z5X), MAO-B (PDB ID: 2V5Z) with their native ligands ADP (and Inhibitor: 6-Bromoindirubin-3′-Oxime; PubChem ID: 448949), Tacrine (PubChem ID: 1935), 5,7-Dichlorokynurenic Acid (PubChem ID: 1779), Harmine (PubChem ID: 5280953) and Safinamide (PubChem ID:131682), respectively, were retrieved from the Protein Data Bank database. Each target was prepared using the AutoDock 4.2 tool, which included solvation parameters, polar hydrogen and Kollman united atom charges and converted the target into pdbqt format.

#### 4.2.2. Ligand Preparation

The three-dimensional structure of ligands was retrieved from the PubChem database. OpenBabel tool (http://openbabel.org (accessed on 21 December 2021)) was used to convert each ligand into pdbqt format [57].

#### 4.2.3. Molecular Docking

Docking studies were carried out using AutoDock Vina [58]. For each target on the specified active site, a grid box was established before docking. The grid configurations of x, y, and z coordinates were set to 60 × 60 × 60, with the grid box’s center set to x: 20.33, y: 16.53 z: −10.38 for GSK3β. The center of x: 4.468, y: 70.009, z: 65.930 for AChE was selected through discovery studio visualizer (BIOVIA) from the attributes of docked native ligand in its specific target protein, while the center was set as x: 5.61, y: 38, z: −17.16 for NMDA [14,59]. The grid configurations of x, y, and z coordinates were set to 33 × 33 × 33 with the grid box center set to x: 140.117, y: 122.247, z: 38.986 for BuChE [60]. Meanwhile, the grid configurations of x, y, and z coordinates were set to 30 × 30 × 30 with grid box center set as x: 40.54, y: 27.04, z: −14.49 for MAO-A and x: 52.34, y: 156.14, z: 28.34 for MAO-B as discussed in previous reports [14,61]. The AutoDock Vina results are displayed as an affinity (kcal/mol). For each ligand–target interaction, the algorithm is divided into 10 modes in descending order. With the PyMol tool’s aid, the best conformation with the lowest energy was chosen and saved as a complex. However, using Discovery Studio Visualizer, the complex produced was further visualized and analyzed (BIOVIA, San Diego, CA, USA, Dassault Systèmes, Velizy, France) [60]. The binding affinity (*Kd*) of ligands for the target protein was calculated from the Gibb’s free binding energy (Δ*G*) using the following relation:$$\Delta G=-RT\;lnK_{d}$$

In this equation, the universal gas constant is denoted as “*R*”, whereas temperature is defined as “*T*”.

The ligands with the minimum Gibb’s free binding energy were selected for further analysis.

### 4.3. Molecular Dynamic (MD) Simulation Study

The stability and dynamic nature of the ligand inside a protein’s binding pocked were determined by performing MD simulation for 100 ns using Desmond (Schrödinger, LLC, New York, NY, USA), as described previously [62]. The initial structures of the protein–ligand complex for MD simulation were obtained from the docking pose. Protein–ligand complex was prepared using Protein Preparation wizard in the Maestro module, which included complex optimization and minimization. The system creator tool prepared all structures within the orthorhombic simulation box, which was then solvated with TIP3P explicit water molecules. The models were neutralized by inserting appropriate counter ions and 0.15 M salt (NaCl) for the physiological condition simulation. LSSB-2005 was used to balance the system with 2000 iterating steps, and forcefield was used to maintain the convergence 1 kcal/mol/Å criteria. MD simulation was performed for 100 ns with an NPT set at 300 K temperature with 1 atm pressure. A Nose-Hoover thermostat and Matrtyna-Tobias-Klein barostat were used to maintain these parameters [63,64]. During simulation, a time step of 2 fs was established, and the energy/structures were recorded every 10 ps. To establish protein-lining complex stability, the metrics like Root-Medium-Square Deviation (RMSD), Root-Medium-Square-Fluctuation (RMSF), radius of gyration (Rg) and Solvent-Accessible Surface Area (SASA) were examined. 

## 5. Conclusions

The present study conducted a screening of potent abrineurin/BDNF natural inducers to find multi-targeting agents against Alzheimer’s Disease. After rigorous initial in silico screening via physicochemical properties, the Lipinski rule, total polar surface area, toxicity assessment, and oral absorption efficiency, five compounds (Soy-B, Huperzine A, Hydroxytyrosol, Alpinetin and Calycosin) were selected. Further molecular docking and interaction analysis against different Alzheimer’s targets (AChE, BuChE MAO-A, MAO-B, GS3Kβ and NMDA) predicted Soy-B to be the most promising compound. However, the stability of Soy-B binding with the target proteins was established by molecular dynamic simulation. Simulation analysis revealed that Soy-B interaction with the active site amino acid residues of the target proteins retained for most of the simulation time, which forms a stable complex. Further in vitro and in vivo explorations are needed to establish Soy-B as a therapeutic candidate for Alzheimer’s disease. However, it is noteworthy to mention that BDNF induction could effectively mitigate neurological ailments, and exploring a solution on BDNF inducer scaffolds for Alzheimer’s appears reasonable. Nevertheless, the present preliminary study predicted that the Soy-B scaffold has all of the characteristics needed for multi-targeting solutions against complex disease like Alzheimer’s.

## Figures and Tables

**Figure 1 entropy-24-00593-f001:**
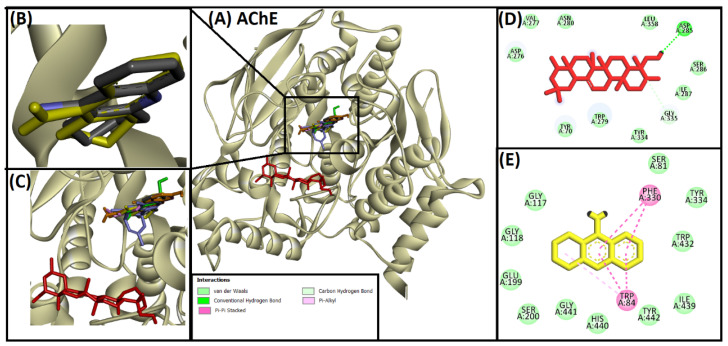
Superimposed image of docked ligands in the active site of AChE (PDB ID: 1ACJ). (**A**) All of the docked ligands (native ligand: grey color; redocked ligand: yellow color; Alpinetin: pink color; Calycosin: golden color; HuperzineA: blue color; Hydroxytyrosol: green color; Soyasapogenol B: red color) in the catalytic active site. (**B**) Superimposed zoomed-in image of native ligand and redocked native ligand. (**C**) Zoomed-in image of all of the docked ligands. (**D**) Molecular interaction analysis of Soyasapogenol B with amino acid residues. (**E**) Molecular interaction analysis of redocked native ligand with amino acid residues.

**Figure 2 entropy-24-00593-f002:**
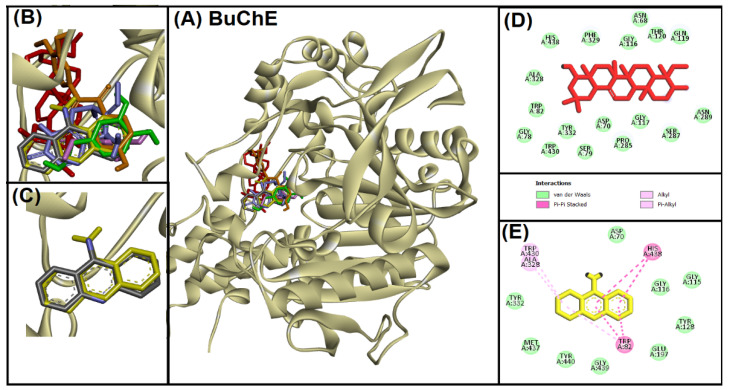
Superimposed image of docked ligands in the active site of BuChE (PDB ID: 4BDS). (**A**) All of the docked ligands (native ligand: grey color; redocked ligand: yellow color; Alpinetin: pink color; Calycosin: golden color; HuperzineA: blue color; Hydroxytyrosol: green color; Soyasapogenol B: red color) in the catalytic active site. (**B**) Zoomed-in image of all of the docked ligands. (**C**) Superimposed zoomed-in image of native ligand and redocked native ligand. (**D**) Molecular interaction analysis of Soyasapogenol B with amino acid residues. (**E**) Molecular interaction analysis of redocked native ligand with amino acid residues.

**Figure 3 entropy-24-00593-f003:**
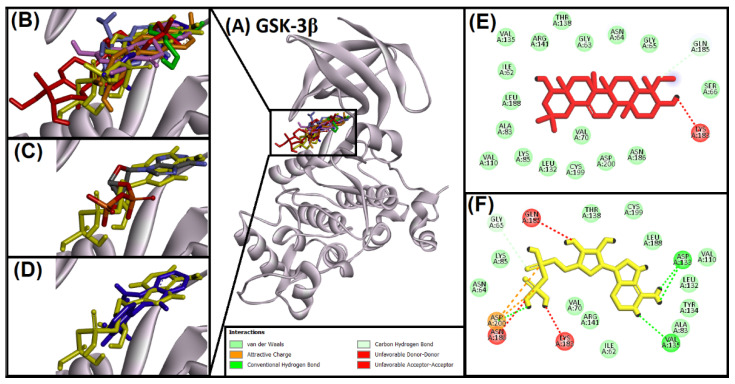
Superimposed image of docked ligands in the active site of GS3Kβ (PDB ID: 1J1C). (**A**) All of the docked ligands (native ligand: grey color; redocked ligand: yellow color; Alpinetin: pink color; Calycosin: golden color; HuperzineA: blue color; Hydroxytyrosol: green color; Bromoindirubin: dark blue; Soyasapogenol B: red color) in the catalytic active site. (**B**) Zoomed-in image of all of the docked ligands. (**C**) Superimposed zoomed-in image of native ligand and redocked native ligand. (**D**) Superimposed zoomed-in image of inhibitor (Bromoindirubin) and native ligand (ADP). (**E**) Molecular interaction analysis of Soyasapogenol B with amino acid residues. (**F**) Molecular interaction analysis of redocked native ligand with amino acid residues.

**Figure 4 entropy-24-00593-f004:**
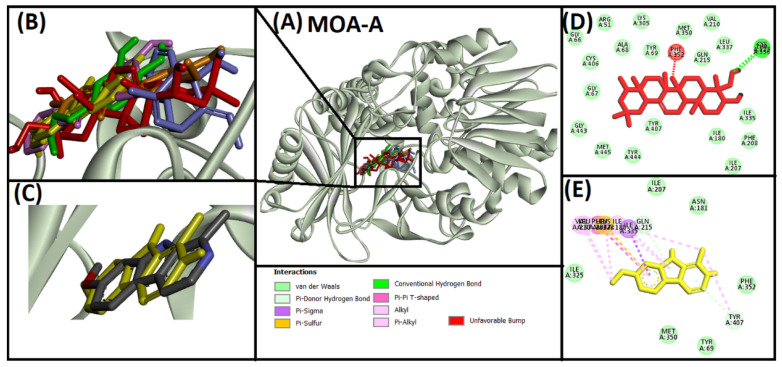
Superimposed image of docked ligands in the active site of MOA-A (PDB ID: 2Z5X). (**A**) All of the docked ligands (native ligand: grey color; redocked ligand: yellow color; Alpinetin: pink color; Calycosin: golden color; HuperzineA: blue color; Hydroxytyrosol: green color; Soyasapogenol B: red color) in the catalytic active site. (**B**) Zoomed-in image of all of the docked ligands. (**C**) Superimposed zoomed-in image of native ligand and redocked native ligand. (**D**) Molecular interaction analysis of Soyasapogenol B with amino acid residues. (**E**) Molecular interaction analysis of redocked native ligand with amino acid residues.

**Figure 5 entropy-24-00593-f005:**
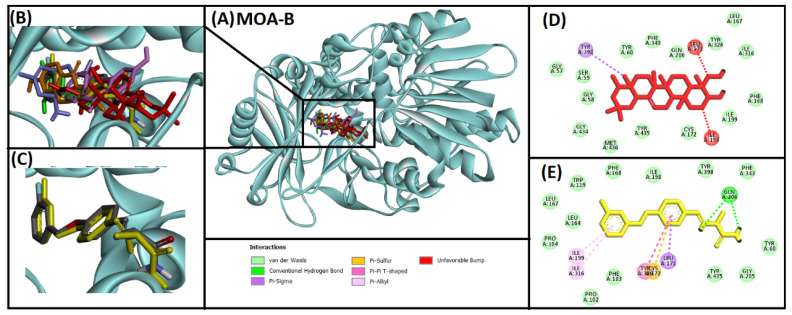
Superimposed image of docked ligands in the active site of MOA-B (PDB ID: 2V5Z). (**A**) All of the docked ligands (native ligand: grey color; redocked ligand: yellow color; Alpinetin: pink color; Calycosin: golden color; HuperzineA: blue color; Hydroxytyrosol: green color; Soyasapogenol B: red color) in the catalytic active site. (**B**) Zoomed-in image of all of the docked ligands. (**C**) Superimposed zoomed-in image of native ligand and redocked native ligand. (**D**) Molecular interaction analysis of Soyasapogenol B with amino acid residues. (**E**) Molecular interaction analysis of redocked native ligand with amino acid residues.

**Figure 6 entropy-24-00593-f006:**
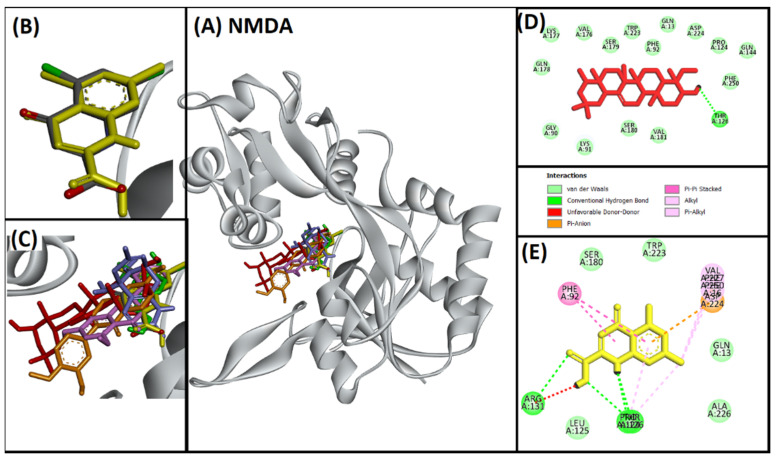
Superimposed image of docked ligands in the active site of NMDA (PDB ID: 1PBQ). (**A**) All of the docked ligands (native ligand: grey color; redocked ligand: yellow color; Alpinetin: pink color; Calycosin: golden color; HuperzineA: blue color; Hydroxytyrosol: green color; Soyasapogenol B: red color) in the catalytic active site. (**B**) Superimposed zoomed-in image of native ligand and redocked native ligand. (**C**) Zoomed-in image of all of the docked ligands. (**D**) Molecular interaction analysis of Soyasapogenol B with amino acid residues. (**E**) Molecular interaction analysis of redocked native ligand with amino acid residues.

**Figure 7 entropy-24-00593-f007:**
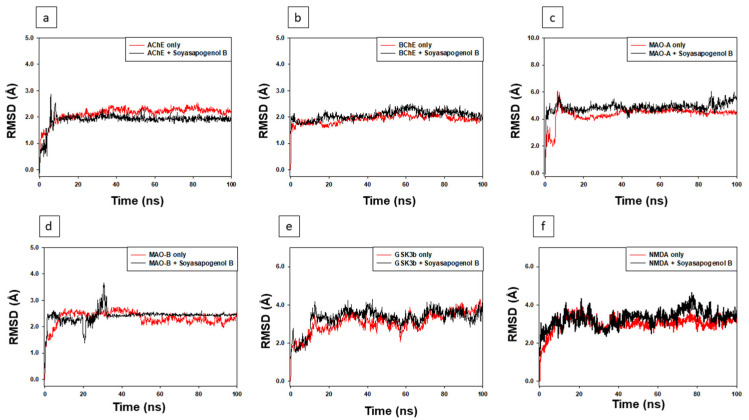
RMSD analysis of Soyasapogenol B with the target proteins. (**a**) AChE, (**b**) BuChE, (**c**) MAO-A, (**d**) MAO-B, (**e**) GS3Kβ, (**f**) NMDA.

**Figure 8 entropy-24-00593-f008:**
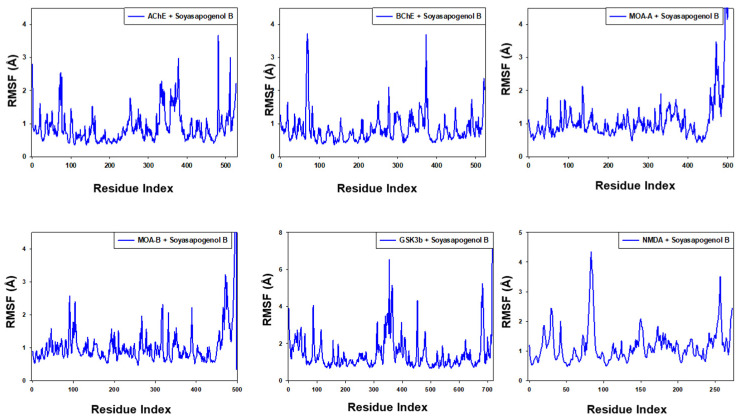
Root mean square fluctuation (RMSF) analysis of Soyasapogenol B with targeted proteins.

**Figure 9 entropy-24-00593-f009:**
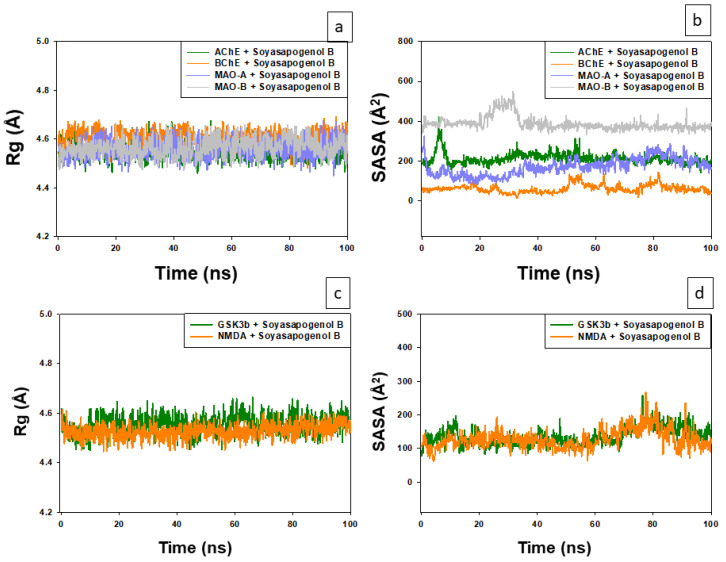
Analysis of radius of gyration (Rg) and solvent accessible surface area (SASA) of Soyasapogenol B. (**a**) Rg of Soy-B with AChE, BChE, MAO-A and MAO-B; (**b**) SASA of Soy-B with AChE, BChE, MAO-A and MAO-B; (**c**) Rg of Soy-B with GS3Kβ and NMDA; (**d**) SASA of Soy-B with GS3Kβ and NMDA.

**Figure 10 entropy-24-00593-f010:**
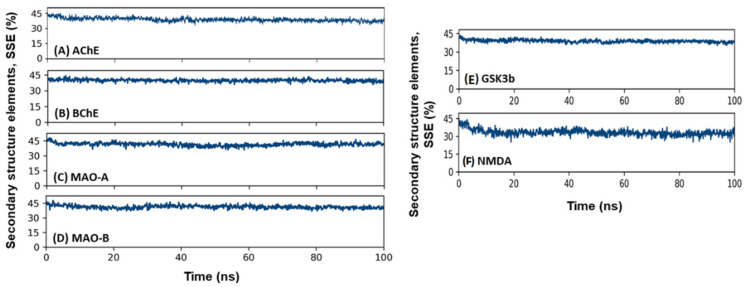
Secondary structure analysis. (**A**) Soy-B + AChE; (**B**) Soy-B + BChE; (**C**) Soy-B + MAO-A; (**D**) Soy-B + MAO-B; (**E**) Soy-B + GS3Kβ; (**F**) Soy-B + NMDA.

**Figure 11 entropy-24-00593-f011:**
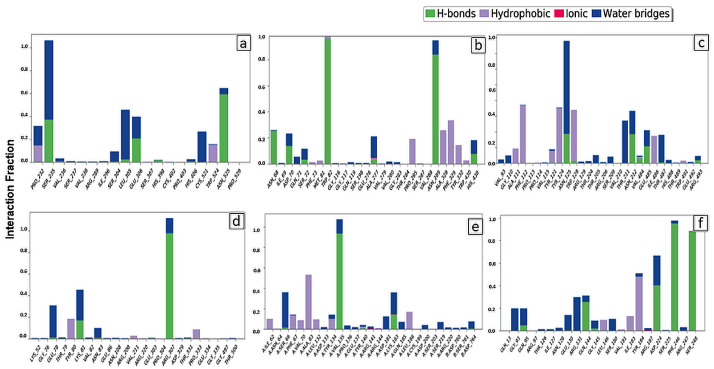
Interaction between Soyasapogenol B with the target proteins. (**a**) Soy-B + AChE; (**b**) Soy-B + BChE; (**c**) Soy-B + MAO-A; (**d**) Soy-B + MAO-B; (**e**) Soy-B + GS3Kβ; (**f**) Soy-B + NMDA.

**Figure 12 entropy-24-00593-f012:**
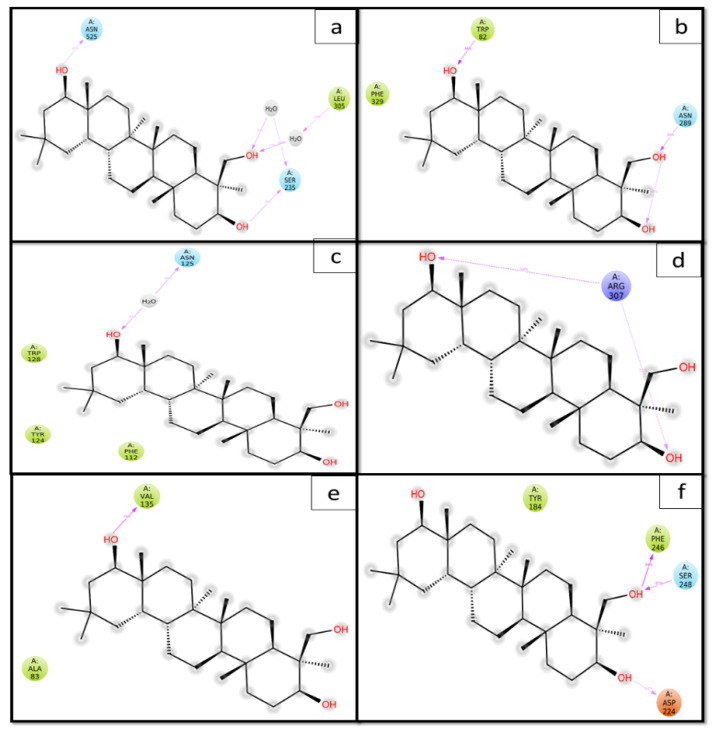
2D representation of the Soyasapogenol B with protein complexes. (**a**) Soy-B + AChE; (**b**) Soy-B + BChE; (**c**) Soy-B + MAO-A; (**d**) Soy-B + MAO-B; (**e**) Soy-B + GS3Kβ; (**f**) Soy-B + NMDA.

**Figure 13 entropy-24-00593-f013:**
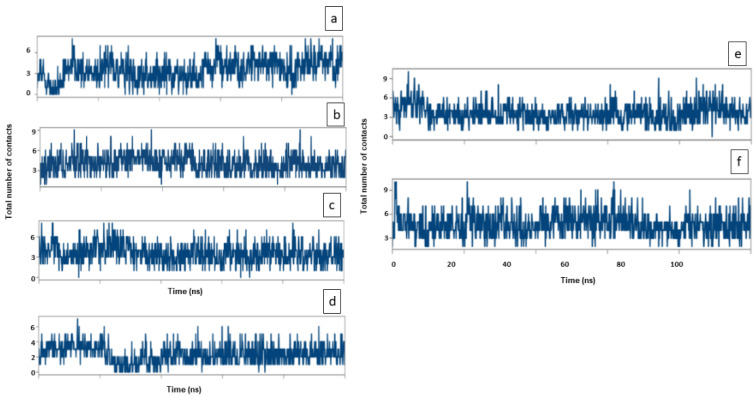
Total number of contacts between Soyasapogenol B and target proteins (**a**) Soy-B + AChE; (**b**) Soy-B + BChE; (**c**) Soy-B + MAO-A; (**d**) Soy-B + MAO-B; (**e**) Soy-B + GS3Kβ; (**f**) Soy-B + NMDA.

**Table 1 entropy-24-00593-t001:** Molecular docking score ΔG (Gibbs free binding energy) and binding affinity (Kd) of selected natural BDNF inducers against neuro-target proteins.

S. No.	Targets	Natural Compounds	Alpinetin	Calycosin	HuperzineA	Hydroxytyrosol	Soyasapogenol B	Native Ligand
1	**AChE**	ΔG (kcal M^−1^)	−7.1	−7.4	−7	−6.7	−8.7	−7.4
		Kd (M^−1^)	1.60 × 10^5^	2.66 × 10^5^	1.35 × 10^5^	8.15 × 10^4^	2.38 × 10^6^	2.66 × 10^5^
2	**BChE**	ΔG (kcal M^−1^)	−8.9	−8.5	−8.7	−6.1	−9.9	−8
		Kd (M^−1^)	3.34 × 10^6^	1.70 × 10^6^	2.38 × 10^6^	2.96 × 10^4^	1.80 × 10^7^	7.31 × 10^5^
3	**GSK**	ΔG (kcal M^−1^)	−8.5	−8.3	−6.9	−6.3	−8.7	−6.7
		Kd (M^−1^)	1.70 × 10^6^	1.21 × 10^6^	1.14 × 10^5^	4.14 × 10^4^	2.38 × 10^6^	8.15 × 10^4^
4	**MOAA**	ΔG (kcal M^−1^)	−8.2	−8.2	−6	−5.9	−7.9	−6.5
		Kd (M^−1^)	1.02 × 10^6^	1.02 × 10^6^	2.50 × 10^4^	2.11 × 10^4^	6.17 × 10^5^	5.81 × 10^4^
5	**MOAB**	ΔG (kcal M^−1^)	−7.1	−7.3	−6.4	−6.3	−7.6	−5.8
		Kd (M^−1^)	1.60 × 10^5^	2.24 × 10^5^	4.91 × 10^4^	4.14 × 10^4^	3.72 × 10^5^	1.78 × 10^4^
6	**NDMA**	ΔG (kcal M^−1^)	−8.3	−7.5	−8.7	−5.5	−8.4	−6.9
		Kd (M^−1^)	1.21 × 10^6^	3.14 × 10^5^	2.38 × 10^6^	1.07 × 10^4^	1.43 × 10^6^	1.14 × 10^5^

## Data Availability

Not applicable.

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
