# Peer review of "Soyasapogenol-B as a Potential Multitarget Therapeutic Agent for Neurodegenerative Disorders: Molecular Docking and Dynamics Study"

_entropy, 2022, doi:10.3390/e24050593_

Round 1
Reviewer 1 Report
Presented theoretical analysis are very interesting but conclusions that this compound could be used as therapeutic molecule in Alzheimer disease are too far. There need to be more results to conclude that something have therapeutic potential and they should include not only stimulation but also in vitro and in vivo study.
Author Response
First of all, we appreciate the time spend by the honorable reviewer from the busy schedule to improve the quality of our MS. All the suggestions given by the reviewers have been addressed and highlighted in Yellow.
Reviewer 1:
- Presented theoretical analysis are very interesting but conclusions that this compound could be used as therapeutic molecule in Alzheimer disease are too far. There need to be more results to conclude that something have therapeutic potential and they should include not only stimulation but also in vitro and in vivo study.
Reply: We agree with the honorable reviewer, in vitro and in vivo studies are warranted to validate the findings before jumping into any conclusion. We have duly modified the sentences in the conclusion section to make it clear. In addition, we have added a statement ꞌThe limitation of the present study is that our findings are only based on the in-silico approach, which need to be further confirmed via in-vitro and in-vivo experimentsꞌ at the end of Discussion section.
Most humbly we would like to state that we have submitted the MS in special issue ꞌMolecular Dynamics Simulations of Biomoleculesꞌ considering that our work is in silico based. It has been observed that in silico prediction often correlates well with experimental findings. Now-a-days, in silico or computational tools are considered as preliminary screening approach before going for any experimental validation. We are currently working on validating the results, and designing more potent Soy-B derivatives by using pharmaceutical chemistry approach based on our in silico preliminary findings.

Reviewer 2 Report
In the present work, Iqbal et al. have analyzed the molecule soyasapogenol-B as a potent multitarget therapeutic agent for neurodegenerative disorders, using a molecular docking and dynamics approach. The work is technical sound and the authors utilized appropriate techniques for the simulation and to verify the results. There are some typewriting errors and some sentences are rambling. However, the following suggestions are recommended:
-In the last paragraph of the introduction, the Author needs to more clearly state the novelty of this paper together with future prospects of this study.
-Authors need to follow the journal format fully in the case of the Reference list. For example, Journal abbreviations, heading, and subheadings etc.
-In the result and discussion section, the author needs to pay more attention and validate their findings with recent previous results and compare if possible.
- The conclusion section must be improved to better explain the obtained results and their potentiality
Author Response
First of all, we appreciate the time spend by the honorable reviewer from the busy schedule to improve the quality of our MS. All the suggestions given by the reviewers have been addressed and highlighted in Yellow.
Reviewer 2:
- In the last paragraph of the introduction, the Author needs to more clearly state the novelty of this paper together with future prospects of this study.
Reply: As per the suggestion of the honorable reviewer, novelty and future prospects of the MS has been duly added in the last paragraph of introduction.
- Authors need to follow the journal format fully in the case of the Reference list. For example, Journal abbreviations, heading, and subheadings etc.
Reply: The reference section has been modified according to the journal style.
- In the result and discussion section, the author needs to pay more attention and validate their findings with recent previous results and compare if possible.
Reply: As per the suggestion of the honorable reviewer, discussion section has been modified to justify the findings through previous reports.
- The conclusion section must be improved to better explain the obtained results and their potentiality.
Reply: As per the suggestion of the honorable reviewer, conclusion has been duly modified for the better explanation of the results.

Reviewer 3 Report
I very much apologize with the Authors and the Editor/Editorial Staff for my delay. In addition to my committments it was a problem with the online system which delayied the download of the manuscript hence the submission. The Authors actually did a lot of work well carried out. As stated in the above fields the manucript, although not highly original, is well prepared and the methods/data are sound. The english language needs revision (some corrections are noted as ticky notes in the attached .pdf). Additional references (see manuscripts by Ramsay et al also recently published, as MOLECULES 2020) should be included . Obviously the manuscript would benefit and become really a greatest work if pharmacological data are included in. This referee would be very pleased to review again or read a version of the manuscript including pharmacological data.
Based on the manuscript itself I selected the item "Accept after minor revision" but I leave the editor the final decision based on thrusting or not a manuscript only based on in silico data.

Author Response
First of all, we appreciate the time spend by the honorable reviewer from the busy schedule to improve the quality of our MS. All the suggestions given by the reviewers have been addressed and highlighted in Yellow.
Reviewer 3:
- The english language needs revision (some corrections are noted as sticky notes in the attached .pdf).
Reply: As per the suggestion of the honorable reviewer, the MS has been duly revised for errors in English language, and corrections suggested in sticky notes have been done.
- Additional references (see manuscripts by Ramsay et al also recently published, as MOLECULES 2020) should be included.
Reply: As per the suggestion of the honorable reviewer, the reference has been added in the revised MS.
- Obviously the manuscript would benefit and become really a greatest work if pharmacological data are included in. This referee would be very pleased to review again or read a version of the manuscript including pharmacological data.
Reply: We appreciate the concern of the honorable reviewer; we are currently working on set-up for experimental validation of the results. As you are aware that
in silico or computational tools are considered as preliminary screening approach before going for any experimental validation. We have only screened the targeted compound Soy-B through the present study, and our next aim is to correlate it with wetlab findings and designed potent inhibitors based on Soy-B scaffold. In fact, we have started working on it. As of now, we submitted our article in in silico special issue ꞌMolecular Dynamics Simulations of Biomoleculesꞌ considering that our work is based on computational screening. Most humbly, we request you to consider it as preliminary in silico screening work.
